# Knowledge and understanding of cardiovascular disease risk factors in Sierra Leone: a qualitative study of patients' and community leaders' perceptions

Agnieszka Ignatowicz,[1] Maria Lisa Odland ![ORCID] ,[1] Tahir Bockarie,[2] Haja Wurie,[3] Rashid Ansumana,[4,5] Ann H Kelly,[6] Chris Willott ![ORCID] ,[7] Miles Witham ![ORCID] ,[8,9] Justine Davies[1,7,10]

AI and MLO are joint first authors.

For numbered affiliations see end of article.

**Correspondence to**
Dr Maria Lisa Odland;
m.l.odland@bham.ac.uk

## ABSTRACT

**Objectives** Prevalence of cardiovascular disease risk factors (CVDRF) is increasing, especially in low-income countries. In Sierra Leone, there are no previous studies on the knowledge and the awareness of these conditions in the community. This study aimed to explore the knowledge and understanding of CVDRF, as well as the perceptions of the barriers and facilitators to accessing care for these conditions, among patients and community leaders in Sierra Leone.

**Design** Qualitative study employing semistructured interviews and focus group discussions.

**Setting** Urban and rural Bo District, Sierra Leone.

**Participants** Interviews with a purposive sample of 37 patients and two focus groups with six to nine community leaders.

**Results** While participants possessed general knowledge of their conditions, the level and complexity of this knowledge varied widely. There were clear gaps in knowledge regarding the coexistence of CVDRF and their consequences, as well as the link between behavioural factors and CVDRF. An overarching theme from the data was the need to create an understanding and awareness of CVDRF in the community in order to prevent and improve management of these conditions. Cost was also seen as a major barrier to accessing care for CVDRFs.

**Conclusions** The knowledge gaps identified in this study highlight the need to design strategies and interventions that improve knowledge and recognition of CVDRF in the community. Interventions should specifically consider how to develop and enhance awareness about CVDRF and their consequences. They should also consider how patients seek help and where they access it.

## INTRODUCTION

Sierra Leone, situated in West-Africa on the Atlantic Ocean, is one of the least developed countries in the world, with a Human Development Index of 0.419 (184 of 189 countries).[1 2] The civil war from 1991 to 2002 disrupted development of the country,

## Strengths and limitations of this study

► This is one of the first studies exploring patient and community knowledge and understanding of cardiovascular disease risk factors in Sierra Leone.
► Interviews and focus group discussions were undertaken by a researcher who is a Sierra Leonean and trained in qualitative research.
► The study focused on people who had sought care for their conditions and only a small number of community leaders. Therefore, the results may not be representative of the entire community.
► The study was done in one district of Sierra Leone.

including health system infrastructure and services, and the Ebola virus disease in 2013–2016 further inhibited development and health service delivery.[3] However, the country is now recovering from these development challenges, with economic growth and moves towards urbanisation and away from a rural subsistence economy.[4] Economic development and urbanisation are likely to be followed by an increase in behavioural risk factors for cardiovascular disease (CVD) such as smoking, high salt and sugar intake, alcohol consumption and low levels of exercise. Like in many other sub-Saharan countries, the epidemic of CVDs in Sierra Leone is likely to be a real and growing problem[5 6]; more than three quarters of CVD related deaths occur in lower and middle-income countries (LMIC).[6] We and others have found that Sierra Leone is facing an increasing burden of cardiovascular disease risk factors (CVDRF), like diabetes and hypertension,[7–9] and the country is faced with the challenge of dealing with these risk factors and their consequences while lacking resources, infrastructure or guidelines.[10–12]

However, despite the clear magnitude of the problem, there is little information about knowledge and understanding of CVDRF among patients and in the community in Sierra Leone, and indeed, in sub-Saharan Africa more broadly.[13–15]

Individual and community awareness and knowledge of CVDRF can lead to better prevention and control of these diseases, as knowledge empowers individuals and their communities to act to prevent or manage these conditions.[16–21] To enable Sierra Leone to deal effectively with an increasing burden of CVDRF, a better understanding of the patient perceptions of and experience with accessing care for CVDRF can help inform the development of interventions that manage and prevent these conditions. Contextually appropriate solutions, capable of grasping the socioeconomic reality of rural life in Sierra Leone, for effective prevention and management of CVDRF are urgently needed. Research on the relationship between community understanding of CVDRF and access to care is therefore key in designing and implementing evidence-based, cost-effective and equitable interventions. In this analysis, we seek to explore the perspectives of patients and community leaders on CVDRF. Building on the findings from a quantitative component of the study,[9] this paper describes knowledge, understanding and practice around CVDRF and perceptions of barriers and facilitators to accessing care.

## METHODS
### Study setting
This study was conducted in Bo District, located in the Southern Province of Sierra Leone. The district has rural and urban areas including the country's second largest city after the capital city, Freetown.[4] The distribution of the population is fairly similar to the larger Sierra-Leonean population considering demographic, socioeconomic, and geographical factors like sex, fertility, urban versus rural population, and employment rate.[4] In Bo, Mende is the most common language spoken, but Krio and English are also used. In the most recent census from 2015, there were 575 478 people living in rural areas (66.1%) and 380 307 (33.9%) living in urban areas, mostly Bo City.[4] 17.4% of the population is over 40 years of age.

### Sampling and participants
Participants were either patients known to have CVDs or risk factors or local community leaders. Patients were interviewed individually, to maximise opportunity for sharing of experiences and perceptions. Community leaders took part in focus group discussions as hierarchical barriers to discussions were felt to be less prominent in community leaders than patients.

We identified patient–participants through a cross-sectional household survey conducted in September–November 2018, which formed the quantitative component of this project.[9] They were selected from those who had answered positively to having CVDRF of hypertension, diabetes, or raised cholesterol, or who indicated they were currently taking medicine to treat or prevent heart disease. To achieve equal representation by sex or rural or urban habitation, we grouped patient–participants by these variables before randomly selecting potential interviewees. Participants were only selected if they had previously indicated were happy to be approached for the qualitative interviews. Analysis was done iteratively and interviews were conducted until saturation in themes was reached.

For focus groups with community leaders, community elders/chiefs were approached through local contacts. Snowball sampling was then used to approach participants from among the business, religious and traditional healer communities across Bo District. Community members consisted of religious leaders such as imams and pastors, town criers and elders, with six to nine members present during each focus group. Two focus groups, one in an urban and one in a rural area, were conducted.

### Data collection
The topic guide was developed based on authors' experiences and knowledge of the literature. The questions were designed to explore patients' perceptions of CVDRF, particularly hypertension, diabetes and their consequences such as stroke or heart attacks, and lived experiences of seeking care and treatment. The interviewer was free to explore various themes that emerged in each individual's interview as well to bring out themes that had emerged during previous interviews. The topic guide was modified for use in the focus group discussions based on the emerging themes from the interviews with patients. The interviews and focus groups were carried out in either Mende or Krio, depending on the language spoken by participants. The interviewers were native Mende or Krio speakers who were trained in qualitative methodologies. Interviews and focus groups were recorded, transcribed and checked by TB and another local transcriber. The interviews and focus group discussions lasted from 45 min to 1½ hours and 1½ to 2 hours, respectively.

### Data analysis
Interviews and focus groups were transcribed, anonymised and translated into English by two of the data collectors—one of whom is the coauthor on this paper—for subsequent analysis. The transcripts were uploaded into the NVivo programme for coding and analysis.[22] Data were analysed thematically using constant comparison[23] within a modified framework approach.[24] Codes were generated both inductively, from the data, and deductively, focussing on articulations of understanding of CVDRF and perceived barriers to accessing care. A sample of interview transcripts was read to identify the initial set of codes by two coauthors of this paper. This generated an initial coding framework that was discussed in an analysis meeting between members of the research team and then used to code all remaining interview and focus group transcripts. Codes were gradually built into

**Table 1** Demographic characteristics of patient participants in the individual interviews (n=37)

| Age | Average (SD) | 55.39 (8.52) |
|---|---|---|
| Gender | Male | 16 |
| | Female | 21 |
| Area of living | Rural | 18 |
| | Urban | 19 |
| Language | Mende | 17 |
| | Krio | 18 |
| | Mende and Krio | 2 |

broader categories and final themes through comparison across transcripts, with further discussion among all team members. In reporting the findings, direct quotes from participants that have been translated into English and anonymised were used. The quotes were translated in a way that closely represented what the person said and not to introduce errors into translation.

### Research team

The collaboration between UK-based and Sierra Leonean researchers, and between medical doctors and social scientists with experience of working in low-income countries and treating patients with CVDRFs, was important for gaining an in-depth understanding of the data and to ensure credibility. During the whole process, the researchers were cognisant of and tried to minimise the effects of their role and background could impact on the data collection and the analysis. To enhance transferability of the project, full details on the setting, participants, data collection and qualitative analysis are provided in the methods section.

### Patient and public involvement statement

Participants were not directly involved in planning the study.

### RESULTS

A total of 37 in-depth interviews and two focus groups were undertaken in this study. The demographic characteristics of the study participants are presented in tables 1 and 2 for the in-depth interviews and focus groups, respectively. The mean age for people participating in in-depth interviews was 55.39 (SD 8.52); there was more women (21) than men (16) and a fairly similar distribution between rural versus urban and the language the interviews were conducted in (mende vs krio). In the urban focus group, there was only one female; however, in the rural focus group, there was a fairly similar distribution between men and women.

In summary, while participants possessed some knowledge of their conditions, the level and complexity of this knowledge varied widely. The majority struggled to define CVDRF. Knowledge gaps were found, particularly in relation to coexistence of CVDRF and their consequences. Out-of-pocket costs were seen as a major barrier to accessing care. An overarching theme from the data was the need to create an understanding and awareness of CVDRF in the community. We discuss these findings in more detail below under three predominant themes of: (1) knowledge and understanding; (2) seeking healthcare; and (3) addressing barriers to access and treatment for CVDRF.

### Knowledge and understanding

*Diabetes.* Participants' understandings of diabetes integrated beliefs about the relationship between 'sugar' and 'the body', expressed through the ideas that diabetes was caused by 'too much sugar'.

> When you have too much of sugar in the body, too much sugar is not good for the body. Patient-participant 9, urban area

> The sugar (…), the glucose is too much. I know, because we were using it when playing football. But then they said is not good to be in your system, the sugar, too much sugar (…) the sugar should not be plenty in your system. Patient-participant 1, urban area

When discussing diabetes, however, the majority of patient–participants had very limited understanding of what causes diabetes. It was common for the participants to state that sugar directly causes diabetes.

> It's a sugar sickness (…) whether it is the ordinary sugar that causes this sickness or where the sugar is coming from, I feel it is sugar that causes this disease [diabetes], (…) that is my own understanding. Patient-participant 22, rural area

Participants talked about diet and cutting down sugar as means of preventing and managing diabetes.

> For us in Sierra Leone, rice is our staple food and we have to eat it and they [neighbours] said even the English barbara (sic) rice has enough sugar content in it and we should not eat. (…) when my sugar content is low, I do eat carbohydrate food, fatty food, but I noticed that it is going up, I stopped eating. Patient-participant 17, urban area

**Table 2** Characteristics focus groups with community leaders

| Area of living | Language | Participants | Gender | Number | Age |
|---|---|---|---|---|---|
| Urban | Krio | 6 district chiefs; 1 paramount chief for chiefdom | 6 males, 1 female | 7 | 40–70 years |
| Rural | Krio | 1 community chief for area; 11 community elders | 5 females, 7 males | 12 | 45–80 years |

(…) what to eat and what not to eat, because like sugar, or any food that is sweet, if you continue to eat it, you will develop sugar sickness. If we advise people, and they follow it up, maybe they will escape most of those bad conditions. Patient-participant 21, rural area

In the focus groups, rural community leaders elaborated that diabetes would be explained to the patients by health professionals in the same way as the previous quotes described. While leaders could not readily identify the causes of diabetes, some were able to related diet and heredity with an increased risk of developing diabetes.

You get two ways which you acquire diabetes, one is the induced and the other one hereditary. The induced one is like plenty-plenty sugar which we eat, like the sugar in food which we take in. The hereditary is when you acquire from the parents. Community leader, focus group in urban area

However, only a small number of participants mentioned lifestyle changes and exercise, which can prevent or reverse diabetes.

My eating habits and other things. Exercise, my eating habits, I should control these. Patient-participant 32, urban area

Lifestyle, lack of exercise…We just sit-down, we do not exercise, we do not warm-up in the morning, we do not run, or even after work, all the time we are in a car or we are on top of motor-bike, so all of that way when we do not exercise it is not well. Community leader, focus group in urban area

Even fewer participants talked about obesity as a factor that can cause diabetes.

They [healthcare professionals] said something on diabetes that if you are overweight will lead to diabetes, but there are people that do not have [large] body that have diabetes. Patient-participant 17, urban area

Most participants in this study could not identify the consequences of diabetes or link diabetes with an increased risk of developing a heart attack or stroke.

*Hypertension.* Hypertension was perceived to be synonymous with raised or high blood pressure affecting the body, caused by 'stress' and 'thinking too much', and described in terms of symptoms such as headache, dizziness or 'hurting heart'.

It is stress, like when you have a lot stress so, like for me so, when you are worried and you are not sleeping well and eating (…) It can be headache or when my heart hurts, the pressure can go up when my heart hurts (…) It's all those things. Patient-participant 8, urban area

High blood pressure is when the blood in your system, the way it is flowing, because it is regulating the heartbeat, goes higher than the normal. Patient-participant 10, urban area

These understandings of hypertension were also substantiated by community leaders.

It can be worry, when you worry too much, you trouble yourself too much, your heart becomes spoilt [sic], and it can cause this thing [hypertension]. Community leader, urban focus group

When you have pressure, you have severe headache and when it hurts for long, it may go above 200 or more than that. Community leader, rural focus group

Because stress was viewed as a cause of hypertension, participants talked about stress reduction—worrying less about financial issues, not having to shout at their children or improved sleep—as the main effective way of managing and treating high blood pressure. A small number of participants reported that controlling diet, in particular salt intake, could prevent and help manage the high blood pressure.

We should have control over our diets, that's how we prevent it [high blood pressure]. Patient-participant 15, urban area

If you stop eating salt, you will see the difference. If you always take drugs, the pressure will not affect you at all. Patient-participant 24, rural area

*Consequences of CVDRF.* Despite limited understanding of causes of CVDRF, some patient–participants talked about the complications of high blood pressure as leading to stroke.

What we know is that stroke is caused by high pressure, if you have high blood pressure and you can't control it, it will result in stoke. Patient-participant 28, rural area

They say when the pressure is high, this is what leads to hypertension. They say that is what makes one of the hands, or the side [of the body], the stroke, the stoke affects and causes the other side of the body to not function. Patient-participant 25, rural area

However, the majority understood stroke in terms of how the person having a stroke looks and the symptoms experienced.

I have seen people who have a bent month, foot is dead or hands are dead. I cannot tell you exactly what the cause of stroke is. Patient-participant 22, rural area

Only one patient–participant mentioned hearth disease as consequences of hypertension.

Hypertension leads to heart failure. Yes, when people die, I feel that it is heart failure, it is heart failure. Patient-participant 20, urban area

Community leaders acknowledged the relationship and coexistence of CVDRF. For example, hypertension was linked with stroke or diabetes.

Because diabetes can go with hypertension, it's common when you have hypertension, when the sugar is more in the system, that is what causes high blood pressure and

hypertension. Community leader, focus group in urban area

This awareness was also expressed by one of the patient-participants, who described hypertension as a 'child' of diabetes.

…diabetes, I feel that it is hypertension's 'pikin' [child, the two are linked], it is their 'pikin', because diabetes has given birth to plenty of 'pikin', one is hypertension. Patient-participant 32, urban area

### Seeking healthcare

While participants talked about traditional doctors and medicine, all described seeking and accessing help through the formal healthcare system. Many were of the opinion that diabetes, hypertension or stroke required hospital level care, but recognised that costs associated with transport, consultation, treatment fees and medication were important barriers to seeking help. Those in urban areas preferred to go to healthcare clinics run by foreign healthcare staff because these clinics, in some cases, charged patients only for treatment and medication.

The reason why most people like the [name of the clinic] is because they will not pay for consultation, they will see doctor for free, and when you go you take your card for free. It is the only place that you will see the doctor for free. We thank God for that, because only when you take that paper for ten thousand Leones (approx. £1 sterling), then you will see the doctor. If you have small money at hand, then you will use it to buy your drugs. Patient-participant 36, urban area

Patient–participants in rural areas preferred the nearby public health centres that typically cared for pregnant women, infants and malaria patients, and according to participants, operated a cost-recovery system for some medication. However, the frequency with which medication was out of stock often meant that participants had to seek hospital level care.

When I cannot travel to [government hospital] there is a clinic here, I have to go there for them to test my pressure, but they don't have drugs, they only check my pressure whether it is high or low. Then I have to go to [government hospital]. Patient-participant 24, rural area

Urban participants talked about seeking help in pharmacies that did not charge for consultations and dispensed medication in the quantities that patients were able to afford.

When I cannot go to hospital, I just go pharmacy. When I go there, they can test me. If it is up [blood pressure], they tell me it is up. They ask which medicine I can take, or which I was taking, then I show them. Community leader, urban focus group

If the sickness [is] serious, we will go to hospital, but for me I go to clinic pharmacy, it's the pharmacy that

I will go to if I do not have enough money. Patient-participant 17, urban area

Financial constraints often led patient–participants to delay seeking care. People therefore preferred to go the pharmacy because you were more likely to get the drugs you though you needed, as compared with using money to travel to a hospital, where they ultimately might not provide you with any drugs or treatment. These participants held an opinion that pharmacies were able to give them the drugs that they could afford and provide more caring treatment.

If you do not have enough money to go to government hospital, you go to where they will manage your life, because at the pharmacy, what you have got, is what you will buy, the amount you have is the kind drugs you will buy. Patient-participant 11, urban area

For some patient–participants, the costs of seeking help forced them to take medication infrequently to make it last longer or in some cases to forgo treatment altogether. Community leaders elaborated that for those patients who already could not afford consultation, additional costs of medication and transport added to their distress.

### Addressing barriers to access and treatment for CVD risk factors

Through the analysis of the data, although costs were highlighted as an issue, lack of knowledge and awareness were singled out as one of the greatest barriers to seeking care for CVDRF. There was a lack of understanding of CVDRF consequences, and when and where to seek care. Moreover, there was a lack of awareness on prevention of conditions. Both patient–participants and community leaders felt that creating understanding and awareness in the community was needed. In the focus groups, community leaders acknowledged the need for better education and prevention.

We need education because they say prevention is better than cure, so when we are able to prevent, then we will reduce the sickness in the community. Community leader, rural focus group

There was a perception that building knowledge, skills and positive attitudes about CVDRF should start at school and involve children and young people.

I would like this education to be extended to schools, because when children are small, you can begin to teach them about this. I think that they will be able to protect themselves so that they will not develop this thing. We need more education, let us not leave it with the adults, let us extend it to various schools. Community leader, rural focus group

Participants agreed that efforts should be increased to raise awareness about CVDRF. However, they noted that despite the magnitude of the CVD burden, there was no specific CVD outreach and sensitisation activities in the community.

These type of sickness, we have not yet seen such activities going on in this community. In case of malaria,

they have started supplying the drugs, they advise people to clean their compound, but these other sickness [diabetes, hypertension and stoke], like the one I have now, I have not heard about it. Patient-participant 26, rural area

Community engagement and awareness programmes were seen as important vehicles for increasing knowledge and understanding. Some ideas were mentioned, such as radio, printed information, door-to-door screening and education, and use of public events and forums. Leaders emphasised that any information should be tailored to their community members' needs and communicated in local languages.

They really need to make booklet with pictures so that people will educate themselves, they can do massive public education, awareness rising meetings. Patient-participant 13, urban area

We need the learning more but let me say, whatever happens, maybe one or two people will be among us that will not hear Krio (…). Let there be somebody that can talk our own language here, Mende, so people can understand in our language. Community leader, rural focus group

Both patient–participants and community leaders agreed that addressing poverty was a key facilitator to increasing access to CVDRF care. They felt that better and more timely access was dependent on addressing the broader social determinants of health, but also health system barriers, such as shortages of drugs.

If people had money, if drugs were available at the hospital, then people would go there as a first point. Like with me, because I have hypertension. If I come to the hospital and I get the treatment I need, I will leave with a positive feeling and refer other people I come across with similar problems. Patient-participant 25, rural area

What I experienced, had it not been the help of chief, I might have died. But he was in Freetown, they rang him, then he brought the drugs. The drugs needed were not in the hospital. So, if it happens that you have no money and the hospital has no drugs, you are going to die. Patient-participant 26, rural area

## DISCUSSION

This paper has explored the knowledge of and attitudes towards CVDRF, and perceptions of barriers and facilitators to accessing care among patients and community leaders in Sierra Leone. In line with work in other LMICs, participants struggled to define diabetes, hypertension or stroke.[15 25] Knowledge gaps were found, particularly around the link between these conditions and their consequences, especially for diabetes. Participants seemed to recognise the increase in the prevalence of various CVDRF in their communities, yet only a few were able to ascribe behavioural risk factors such diet and weight management or regular exercise to CVDRF.

There was some knowledge among community leaders regarding the relationship and coexistence of CVDRF. Poverty and lack of medications in health facilities were perceived as factors hindering treatment-seeking behaviour for CVDRF conditions. Economic considerations, in particular, were important when seeking help. This is in line with wider evidence from sub-Saharan Africa, which suggests that widespread poverty affects treatment seeking and management of CVDRF in the community.[25 26] Patient–participants often talked about seeking help in public health facilities and pharmacies that did not charge for consultation, only choosing hospitals when they judged their symptoms to be more serious. A recent study into the community dynamics surrounding noncommunicable diseases (NCDs) in Sierra Leone revealed that seeking help from formal health facilities was often the last resort for many patients.[27] However, our findings also indicated that the lack of awareness was seen as the biggest barrier to accessing care and managing CVDRF for patients and community leaders alike.

These knowledge gaps and perceptions should be important considerations when designing and implementing interventions to enhance knowledge of CVDRF in Sierra Leone. They highlight the need to raise awareness about CVDRF consequences, but also behavioural factors and their potential relationships to the development of CVDRF. The limited understanding of the relationships between behavioural factors, different CVDRF conditions and/or their consequences among patients and community leaders in this study brings to the fore the need to develop community health education and promotion programmes specifically targeting awareness of these links. To put this to context, in the quantitative component of this study, we found hypertension was present in 49.6%, hypercholesterolaemia in 6.7%, and diabetes in 3.5% of the study population. Moreover, 25.6% of the population were current or ex (within the last year) smokers and 77.1% had at least one CVDRF.[9] Addressing the burden of CVD in Sierra Leone will require interventions to achieve better awareness, but also support detection and control of these conditions. A recent systematic review on knowledge of CVD risk in sub-Saharan Africa found that the major sources of information for patients included television, radio, newspapers, healthcare professionals and family members or relatives.[15] Patients and community leaders in our study agreed that implementing and intensifying awareness via radio and other means could be potential platforms to improve knowledge of CVDRF. Additionally, they suggested that the educational programmes could benefit from the development of a more personalised approach tailored to their community members' needs and communicated in local languages. This targeted approach presents an important consideration for development of any educational initiatives. In order to be more successful, it has been suggested that educational programmes should tap into the local views of those directly affected by the CVD conditions.[15] An approach to CVDRF prevention, as suggested by the community leaders in this study, requires these programmes to start at school and be incorporated into curricula, but education about preventing disease alone

may not be sufficient. Basic educational skills are important components for achieving health.[28] Fundamental biological knowledge and skills such as reasoning ability are also necessary. Effective interventions to enhance CVDRF knowledge in schools and beyond, require deep understanding about the relationships between education and health.

Furthermore, there is a need for better policy to change the health system and accompanying resources to enhance the awareness and practices around CVDRF. Most patients in our study had to pay for medical care and medication out of pocket, which constituted a major barrier to accessing care and adherence to long-term or lifelong treatment. Similar findings have been documented in other in low-resourced settings, linking poverty and the lack of policy level interventions to increase access and affordability of services.[28] However, some successful examples of such interventions exist. MoPoTsyo is a community management scheme in that uses a revolving drug fund for CVDRF,[29] and our recent analysis of this programme showed substantial and sustained reductions in blood pressure and glucose for those enrolled.[30] To ensure all healthcare needs of individuals are met in countries with underdeveloped health systems, health system policy interventions could consider adopting more community centric models of care.[18]

What is more, patients and community leaders in our study talked about seeking help in local pharmacies. Since community pharmacists seem to be accessible healthcare providers, they may be in position to provide early detection of CVDRF and provide education and counselling when appropriate. Evidence of pharmacists' interventions and the prevention and management of CVD in primary care, particularly in collaboration with other healthcare professionals, demonstrates a positive effect on various patient outcomes.[27]

Community-level interventions may also be a possible solution for preventing or managing CVDRF, as has been done previously in Bangladesh.[16] In this study, communities were mobilised by applying a participatory learning and action cycle to raise knowledge and awareness of diabetes. Engaging the community can be done more effectively through pre-existing community structures and healthcare providers, alongside deliberate efforts to improve the performance and effectiveness of the healthcare system. Viable community structures exist within Sierra Leone that have the power to deliver better health to community dwellers. There are hierarchical community structures (of chiefs, sub-chiefs and village leaders) with regular community meetings, which could be important sites to enhance health behaviour. Finally, as Sierra Leone is developing an NCD plan and its health system, there needs to be focus on CVDRF prevention, monitoring and control. However, any initiatives or strategies should be supported by regular monitoring and evaluation by the Ministry of Health in order to evaluate the impact on future CVDRF burden to inform and strengthen the future interventions.

There are several limitations in this study. Interviews were only conducted with people who had sought care

for conditions and therefore their knowledge may not be reflective of the general population. Difference in knowledge to the general population may also be found because people with CVDs are likely to be better educated and of higher socioeconomic status than those without these diseases. Explanations and examples were offered to patients to enhance their understanding of the CVDRF. However, some patients were reluctant to attend interviews because of the fear of receiving bad news about their health. Although the interviews were done by an interviewer who was fluent in the local language, some of the concepts that were being discussed may not have had a local language equivalent, this could have led to additional probing and influenced participants' responses. In some instances, and despite the researcher's clarification and probing, participants remained were unable to answer the questions. To reduce the effect of these biases in analysis, the authors of this paper undertook member checks to verify whether the interpretations of results reflect what participants actually intended to say, and organised frequent meetings between all study team members in which developing insights were critically discussed. Finally, given the geographical confines of the study setting, the generalisability of these results to other LMIC contexts may be limited. However, the knowledge gaps identified in this study are consistent with findings from other African countries, despite varied levels of socioeconomic growth and literacy levels. For Sierra Leone, however, the transferability of these findings to other similar groups of patients and community leaders may seem justified given variation within our sample for the study.

## CONCLUSIONS

This study provides insights into how CVDRF are understood in Sierra Leone, and how the understanding and perception affects treatment seeking and management for CVD in the community. Knowledge gaps were found, particularly around understanding of the factors that cause CVDRF and their consequences. A lack of awareness was seen as the biggest barrier to accessing care and managing CVDRF in the community. There is need to design strategies and interventions that improve knowledge and recognition of CVDRF, and use the perspectives of those affected by these conditions. However, addressing the burden of CVDRF in Sierra Leone will require interventions not only to achieve better awareness, but also support detection and control of these conditions.

**Author affiliations**
[1]Institute of Applied Health Research, University of Birmingham, Birmingham, UK
[2]Warwick Medical School, University of Warwick, Coventry, UK
[3]College of Medicine and Allied Health Sciences, University of Sierra Leone, Freetown, Western Area, Sierra Leone
[4]Mercy Hospital Research Laboratory, Bo, Sierra Leone
[5]School of Community Health Sciences, Njala University, Bo, Sierra Leone
[6]Department of Global Health and Social Medicine, King's College London, London, UK

[7]King's Centre for Global Health and Health Partnerships, King's College London, London, UK
[8]Age Research Group, NIHR Newcastle Biomedical Research Centre, Newcastle University, Newcastle upon Tyne, UK
[9]Newcastle Upon Tyne Hospitals NHS Foundation Trust, Newcastle Upon Tyne, UK
[10]Medical Research Council/Wits University Rural Public Health and Health Transitions Research Unit, Faculty of Health Sciences, School of Public Health, University of the Witwatersrand, Johannesburg, South Africa

**Acknowledgements** MDW acknowledges support from the NIHR Newcastle Biomedical Research Centre. Allieu Abu Sheriff DC; Albert Sidikie Sama FM, who helped with the organisation, transcription and translation of the interviews.

**Contributors** JD, AK, CW and MW conceived and designed the overall study. JD, AK, CW, TB, HW and RA coordinated data collection and preparation. AI and MLO conducted the analysis, and wrote, and revised the manuscript. JD supervised the analysis, write up and development of the manuscript. All authors substantively reviewed manuscripts, inputted into revisions, and approved the final manuscript. MW and JD are joint last authors.

**Funding** This work was supported by the Wellcome Trust, grant number 209921/Z/17/Z.

**Competing interests** AI reports grants from the National Institute of Applied Health Research during the conduct of this study.

**Patient and public involvement** Patients and/or the public were not involved in the design, or conduct, or reporting, or dissemination plans of this research.

**Patient consent for publication** Not required.

**Ethics approval** Ethical approval was sought and given from the Sierra Leone Ethical and Scientific Review Committee and the BDM Research Ethics sub-committee at King's College London (HR-17/18-7298). Consent to undertake the study was obtained from each village chief or community leader. Consent was obtained from all individuals participating in the study. In the event where participants were illiterate, the consent form was read out to them in the local language and an inked-thumb signature obtained.

**Provenance and peer review** Not commissioned; externally peer reviewed.

**Data availability statement** Data may be obtained from a third party and are not publicly available. Data are not publicly available as consent was not given by participants for this to take place.

**ORCID iDs**
Maria Lisa Odland http://orcid.org/0000-0003-4340-7145
Chris Willott http://orcid.org/0000-0003-0940-4215
Miles Witham http://orcid.org/0000-0002-1967-0990

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
