## [Reviewer comments · BMJ Open]

ARTICLE DETAILS

TITLE (PROVISIONAL)	Knowledge and understanding of cardiovascular disease risk factors in Sierra Leone: a qualitative study of patients' and community leaders' perceptions
AUTHORS	Ignatowicz, Agnieszka; Odland, Maria Lisa; Bockarie, Tahir; Wurie, Haja; Ansumana, Rashid; Kelly, Ann; Willott, Chris; Witham, Miles; Davies, Justine

VERSION 1 – REVIEW

REVIEWER	Kim Greaves Sunshine Coast University Hospital, Queensland, Australia Research School of Population Health, Australian National University, National Centre for Epidemiology and Population Health, ACT 2601
REVIEW RETURNED	14-Apr-2020

GENERAL COMMENTS	This is a valuable study which is informative and adds to the minimal literature available on CVD risk factor knowledge in Sierra Leone. In this study the authors conduct a number of interviews and focus groups to gain an understanding of the population's perspective and understanding of CVD risk and access to care. They demonstrate that that knowledge is poor, access to healthcare and drugs is a problem as a result of costs to the patient. Comments 1. In the introduction it would help readers understand the scale of the problem in sub-Saharan Africa if more information could be given on CVD risk and CVD prevalence. To do this I suggest providing quantitative data of the growing scale of CVD risk and CVD itself in sub-Saharan Africa. E.g. DM 6.2% prevalence; 37% HT; 34% smoking; 27% 3-5 RF for CVD in WHO STEP; the fact that CVD is growing - an estimated 1 million deaths were attributable to CVD in SSA, constituting 5.5% of all global CVD-related deaths and 11.3% of all deaths in Africa. https://www.ahajournals.org/doi/full/10.1161/CIRCULATIONAHA.118.037367 Kofi Amiga. Also this reference is important too: Roth GA, Forouzanfar MH, Moran AE, Barber R, Nguyen G, Feigin VL, Naghavi M, Mensah GA, Murray CJ. Demographic and epidemiologic drivers of global cardiovascular mortality. N Engl J Med. 2015; 2. Please discuss: How representative was the group interviewed of the general population – noting that they were business owners, imams and seem to have higher socio-economic status? What was the educational status of the patients interviewed? Discuss how groups interviewed were reflective of the general population living in Bo? Please discuss these points as they could influence the results of the study
---

	3. In the discussion it would be valuable to talk more (in a paragraph) about the influence of health education (and lack thereof), and its influence on the ability to understand the meaning of risk factors in CVD. How can a person understand the adverse effects of hypertension or diabetes in CVD, if they do not understand the basics rudiments on how the cardiovascular system works? Please see the importance of health education on public health: https://www.ncbi.nlm.nih.gov/pmc/articles/PMC4691207/ 4. It is not clear to me why the reference is mentioned 'Odland et al. "Prevalence and access to care for cardiovascular risk factors in older people in Sierra Leone: A cross-sectional survey' in the text – is this unpublished – in which case this needs to be made clear. I am not sure you can reference this. Check with authorship guidelines.
--	---

REVIEWER	D. Edmund Anstey Columbia University Irving Medical Center
REVIEW RETURNED	30-Apr-2020

GENERAL COMMENTS	In this manuscript by Ignatowicz, Odland et al, the authors examine the knowledge and the awareness of cardiovascular disease risk factors among patients and study participants in Sierra Leone. Data was gathered through focus groups and interviews among community members and community leaders. The authors identified many knowledge gaps in the community as well as insight into the participants perceived barriers to care. Overall the manuscript was well written. The methodologic approach was appropriate with rigorous effort to ensure proper procedures in a qualitative design and avoid introducing bias. Some comments for consideration: Background The background section is long. Some of the details may be moved to the discussion. Methods: Page 8, Line 47 – Who were the interviewers and do they have prior experience conducting focus groups and was there any training on conducting focus groups? Are they study authors, community members, etc? There is some discussion of this in the Discussion (page 24, line 17-22) which, if applicable to all focus groups/interviews, should be moved to the Methods. Discussion Please provide more discussion about how the knowledge gaps in cardiovascular disease risk factors identified in Sierra Leone relate to those in other areas in Africa or across the globe. This is mentioned (page 2, lines 8-15; page 24, line 47-52) but a longer discussion on this topic may help guide the reader regarding the relative importance of examining this question specifically in this population. Minor Comments: Results: Table 1 – For age, please provide standard deviation or interquartile range if non-normal distribution Page 6, Line 38 – Please spell abbreviation Page 10, Line 24-26 – Please move to methods
---

VERSION 1 – AUTHOR RESPONSE

Reviewer 1:

In the introduction, it would help readers understand the scale of the problem in sub-Saharan Africa if more information could be given on CVD risk and CVD prevalence. To do this I suggest providing quantitative data of the growing scale of CVD risk and CVD itself in sub-Saharan Africa. E.g. DM 6.2% prevalence; 37% HT; 34% smoking; 27% 3-5 RF for CVD in WHO STEP; the fact that CVD is growing - an estimated 1 million deaths were attributable to CVD in SSA, constituting 5.5% of all global CVD-related deaths and 11.3% of all deaths in Africa.

<https://www.ahajournals.org/doi/full/10.1161/CIRCULATIONAHA.118.037367> Kofi Amiga.

Also this reference is important too: Roth GA, Forouzanfar MH, Moran AE, Barber R, Nguyen G, Feigin VL, Naghavi M, Mensah GA, Murray CJ. Demographic and epidemiologic drivers of global cardiovascular mortality. *N Engl J Med*. 2015;

Response:

Thank you for your comment and useful references. We have rewritten the Introduction and added the sentence about the scale of CVD in sub-Saharan Africa and CVD related deaths in LMICs. (p.4, starting from line 102).

We would also like to reference our quantitative paper which is under revision in *BMJ open* as well if the papers are published at the same time. Reference [bmjopen-2020-038520](https://doi.org/10.1136/bmjopen-2020-038520).

Reviewer 1:

Please discuss: How representative was the group interviewed of the general population – noting that they were business owners, imams and seem to have higher socio-economic status? What was the educational status of the patients interviewed? Discuss how groups interviewed were reflective of the general population living in Bo? Please discuss these points as they could influence the results of the study

Response:

We are sorry if this was not clear. As stated in the methods, there were two groups of people who supplied data for this analysis.

1. Community leaders – some of whom were imams, business owners, etc

This is described in the following sentences:

(p.5 line 139-140)

“Participants were either patients known to have cardiovascular diseases or risk factors or local community leaders”

(p.6 line 155-156)

“For focus groups with community leaders, community elders/chiefs were approached through local contacts.”

2. Members of the public who had NCD risk factors. Although these members of the public were selected from the household survey that is presented in the accompanying paper, we conducted the interviews anonymously and are not able to link them back to their socioeconomic details supplied in

the household survey.

We selected these community dwellers by first stratifying the groups by sex and rural or urban habitation, then randomly selecting interviewees from these groups.

We have made this clearer in the manuscript (p.6 line 145-151), as follows:

“We identified patient-participants through a cross-sectional household survey conducted in September-November 2018, which formed the quantitative component of the project. They were selected from those who had answered positively to having CVDRF of hypertension, diabetes, or raised cholesterol, or who indicated they were currently taking medicine to treat or prevent heart disease. To achieve equal representation by sex or rural or urban habitation, we grouped patient-participants by these variables before randomly selecting potential interviewees.”

However, although randomly selected, it is likely that they are of higher socioeconomic status than people without cardiovascular disease, as we have found that risk factors are more prevalent in people of higher socioeconomic status. This is included as a limitation (p.22 line 556-558):

“Difference in knowledge to the general population may also be found because people with cardiovascular diseases are likely to be better educated and of higher socioeconomic status than those without these diseases”

Reviewer 1:

In the discussion it would be valuable to talk more (in a paragraph) about the influence of health education (and lack thereof), and its influence on the ability to understand the meaning of risk factors in CVD. How can a person understand the adverse effects of hypertension or diabetes in CVD, if they do not understand the basics rudiments on how the cardiovascular system works? Please see the importance of health education on public health:

<https://www.ncbi.nlm.nih.gov/pmc/articles/PMC4691207/>

Response:

Thank you – we feel that this is an important point. We have therefore expanded on this in our discussion to acknowledged the need for basic educational skills as well as the need to recognize the link between education/and health (p.20, line 509-517).

“In order to be more successful, it has been suggested that educational programmes should tap into the local views of those directly affected by the CVD conditions. A better approach to CVDRF prevention, as suggested by the community leaders in this study, requires these programmes to start at school and be incorporated into curricula, but education about preventing disease alone may not be sufficient. Basic educational skills are important components for achieving health. Fundamental biological knowledge and skills such as reasoning ability are also necessary. Effective interventions to enhance CVDRF knowledge in schools and beyond, require deep understanding about the relationships between education and health.”

Reviewer 1:

It is not clear to me why the reference is mentioned ‘Odland et al. "Prevalence and access to care for cardiovascular risk factors in older people in Sierra Leone: A cross-sectional survey’ in the text – is this unpublished – in which case this needs to be made clear. I am not sure you can reference this. Check with authorship guidelines.

Response:

Thank you – we have removed the reference.

Given that the papers are inherently linked, we submitted the papers together and are hoping that they will be published together to enable us to refer readers of one easily to the other.

Reviewer 2:

Background

The background section is long. Some of the details may be moved to the discussion.

Response:

Thank you for your comment. We have shortened the Introduction.

Reviewer 2:

Methods

Page 8, Line 47 – Who were the interviewers and do they have prior experience conducting focus groups and was there any training on conducting focus groups? Are they study authors, community members, etc? There is some discussion of this in the Discussion (page 24, line 17-22) which, if applicable to all focus groups/interviews, should be moved to the Methods.

Response:

There were two interviewers doing the study. The main interviewer (author TB), who had been trained in qualitative methods, including conducting focus group discussions did most of the interviews which were in Krio. The interviews that required a Mende speaker were done by TB and an additional interviewer, trained in interview methods by TB, and fluent in Mende. The Mende speaking interviewer only conducted the interviews and didn't contribute to the manuscript development, they are therefore not an author.

We have added the following to the methods (p.7 line 171-172):

“The interviewers were native Mende or Krio speakers who were trained in qualitative methodologies”

We have moved the relevant sentence to the methods.

Reviewer 2:

Discussion

Please provide more discussion about how the knowledge gaps in cardiovascular disease risk factors identified in Sierra Leone relate to those in other areas in Africa or across the globe. This is mentioned (page 2, lines 8-15; page 24, line 47-52) but a longer discussion on this topic may help guide the reader regarding the relative importance of examining this question specifically in this population.

Response:

We have addressed this comment throughout the discussion section, indicating where our findings align with those from previous studies in other LMICs and sub-Saharan Africa (e.g. p.19, line 470, 478-480, and 483-487; p.21 line 522-524).

Reviewer 2:

Minor comments

Results:

Table 1 – For age, please provide standard deviation or interquartile range if non-normal distribution

Page 6, Line 38 – Please spell abbreviation

Page 10, Line 24-26 – Please move to methods

Response:

Thanks for these minor comments. We have made the suggested changes.

VERSION 2 – REVIEW

REVIEWER	Kim Greaves Sunshine Coast University Hospital, Queensland Australia
REVIEW RETURNED	25-Aug-2020

GENERAL COMMENTS	The reviewer completed the checklist but made no further comments.
--

REVIEWER	D. Edmund Anstey Columbia University Medical Center, USA
REVIEW RETURNED	06-Jul-2020

GENERAL COMMENTS	The authors have thoroughly addressed this reviewers concerns. My only comments are minor. Line 100: introduce/define the abbreviation for cardiovascular disease (CVD) at first use and not in line 102 Table 2: Please include precise age ranges instead of "80+" as, as written, it's unclear if these numbers are to represent precise ages or rounded numbers. Given the small sample size, precise numbers may be appropriate.
--